# Human Pose Detection for Robotic-Assisted and Rehabilitation Environments

**Óscar G. Hernández** , **Vicente Morell, José L. Ramon** and **Carlos A. Jara** *

Human Robotics Group, University of Alicante, San Vicente del Raspeig, 03690 Alicante, Spain;
oghernandez@unah.edu.hn (Ó.G.H.); vicente.morell@ua.es (V.M.); jl.ramon@ua.es (J.L.R.)
* Correspondence: carlos.jara@ua.es; Tel.: +34-965903400 (ext. 1094)

**Featured Application: The proposed technology is useful for the estimation of human biomarkers in rehabilitation processes or for any application that needs human pose estimation.**

**Abstract:** Assistance and rehabilitation robotic platforms must have precise sensory systems for human–robot interaction. Therefore, human pose estimation is a current topic of research, especially for the safety of human–robot collaboration and the evaluation of human biomarkers. Within this field of research, the evaluation of the low-cost marker-less human pose estimators of OpenPose and Detectron 2 has received much attention for their diversity of applications, such as surveillance, sports, videogames, and assessment in human motor rehabilitation. This work aimed to evaluate and compare the angles in the elbow and shoulder joints estimated by OpenPose and Detectron 2 during four typical upper-limb rehabilitation exercises: elbow side flexion, elbow flexion, shoulder extension, and shoulder abduction. A setup of two Kinect 2 RGBD cameras was used to obtain the ground truth of the joint and skeleton estimations during the different exercises. Finally, we provided a numerical comparison (RMSE and MAE) among the angle measurements obtained with OpenPose, Detectron 2, and the ground truth. The results showed how OpenPose outperforms Detectron 2 in these types of applications.

**Keywords:** human–robot interaction; human pose estimation; robotic rehabilitation

## 1. Introduction

A high demand of services for assisted and rehabilitation environments is expected from the health status of the world due to the COVID-19 pandemic. Currently, according to the WHO (World Health Organization), existing rehabilitation services have been disrupted in 60–70% of countries due to this pandemic in order to avoid human contact. Therefore, countries must face major challenges to ensure the health of their population. Robotic platforms are a great solution to ensure assistance and rehabilitation for disabled people using human–robot interaction (HRI) capabilities. HRI is currently a topic of research that contributes by means of several research approaches for the physical and/or social interaction of humans and robotic systems [1] in order to achieve a goal together.

The number of people with congenital and/or acquired disabilities are quickly increasing, and therefore, there are many dependents who lack the necessary autonomy for a fully independent life. Among them, stroke is one of the main causes of these acquired disabilities throughout the world. Acquired brain injury (ABI) is a clinical-functional situation triggered by an injury of any origin that acutely affects the brain, causing neurological deterioration, functional loss, and poor quality of life as a result. It can be due to various causes, with stroke and head trauma the most frequent in our environment. Patients with ABI suffer cognitive and motor sequelae. In stroke patients, motor sequelae are usually more severe in the upper limb. In published studies, it has been reported that 30–60% of hemiplegic patients due to a stroke remain with severely affected upper limbs after 6 months of the event, and only 5–20% manage a complete functional recovery.

Physical medicine and rehabilitation are the most important treatment methods in ABI because they help patients reutilize their limbs at maximum capacity. Intensive therapies and repetitive task-based exercises are very effective treatments for motor skills recovery [2]. One of the most important processes of physical therapy requires manual exercises, in which the physiotherapist and the patient must have one-to-one interaction. The goal of the physiotherapist in this process is to help the patients achieve a normal standard of range of motion in their limbs and to strengthen their muscles. Rehabilitation robotic platforms pursue the recovery of impaired motor function. The majority of rehabilitation robotics research to date has focused on passive post-stroke exercises (e.g., [3,4]). The use of assistive robotics in rehabilitation allows the assistance of the physiotherapist in certain exercises that require repeated movements with high precision. The robot can fulfil the requirements of the cyclic movements in rehabilitation. Additionally, robots can successfully control the forces applied and can monitor the therapy results objectively by using their sensors.

Robots intended for upper limb rehabilitation can accomplish active and passive bilateral and unilateral motor skills training for the wrist, forearm, and shoulder. MIT-Manus is one of the most well-known upper limb rehabilitation robots [5]. It was developed for unilateral shoulder or elbow rehabilitation. MIME is another well-known upper limb rehabilitation robot, developed for elbow rehabilitation using the master–slave concept [6]. The movement of the master side of the robot is reproduced on the slave side. The 2-DOF robot can perform flexion–extension and pronation–supination movements. The Assistive Rehabilitation and Measurement (ARM) Guide is a bilateral rehabilitation system for upper limb rehabilitation using an industrial robot [7]. It assists the patient in following a trajectory. It also serves as a basis for the evaluation of several key motor impairments, including abnormal tone, incoordination, and weakness. The GENTLE/s system uses a haptic interface and virtual reality techniques for rehabilitation. The patients can move their limbs in a three-dimensional space with the aid of the robot [8]. The authors of [9] presented a rehabilitation robot with minimum degrees of freedom to train the arm in reaching and manipulation, called reachMAN2. All these previous robotic devices provide the potential for patients to carry out more exercise with limited assistance, and dedicated robotic devices can progressively adapt to the patients' abilities and quantify the improvement of the subject.

Robotic platforms for assistance and rehabilitation must have precise sensory systems for HRI. Therefore, they must recognize human pose or human gestures to improve the performance and safety of human–robot collaboration in these environments [10,11]. Our study sought to obtain a marker-less good pose estimation using a low-cost RGB camera for upper limb robotic rehabilitation environments. We set-up a multiple-RGBD-calibrated-cameras system to measure the goodness of the available methods.

## 2. Human Pose Detection and Body Feature Extraction: A State of the Art

The human body is a very complex system composed of many limbs and joints, and the exact detection of the position of the joints in 2D or 3D is a challenging task [12], as it requires a specific assumption within biomechanics research in robotic rehabilitation environments [13]. In addition, HRI environments are complex and nondeterministic, and it is not easy to ensure the user's safety during interaction with the robot. Currently, this assumption is a research topic in other areas, such as Industry 4.0 [14,15]. The resolution of this issue involves constant position tracking, intention estimation, and action prediction of the user. This problem can be faced by a proper sensory system. On the one hand, some contributions employ inertial measurement units (IMUs) for motion capture, especially in medical applications and motor rehabilitation analysis [16,17]. However, this type of sensor requires the correct placement of passive/active markers on the body before each capture session, and they are insufficient for HRI environments.

On the other hand, this issue can also be faced as a computer vision problem, basically using two approaches: marker-based and marker-less. Marker-based approaches,

such as motion capture systems (MoCap), have significant environmental constraints (markers in the human body) and are relatively complex, expensive, and difficult to maintain. Marker-less approaches have fewer environmental constraints, and they can give a new understanding about human movements [18]. This issue requires the processing of complex information to develop an algorithm that recognizes human poses or skeletons from images. Therefore, an easy-to-use marker-less motion capture method is desirable for these robotic rehabilitation environments. In this paper, we analyzed the performance of the estimation of shoulder and elbow angles for the development of rehabilitation exercises using CNN (convolutional neural network)-based human pose estimation methods.

There is extensive research about marker-less approaches for human tracking motion. In these approaches, depth cameras such as Kinect (RGB-D) provide additional information about the 3D shape of the scene. Kinect has become an important 3D sensor, and it has received a lot of attention thanks to its rapid human pose recognition system. Its low-cost, reliability, and rapid measurement characteristics have made the Kinect the primary 3D measuring device for indoor robotics, 3D scene reconstruction, and object recognition [19]. Several approaches for real-time human pose recognition have been performed only using a single sensor [20–22], but it can have substantial errors with partial occlusions.

In recent years, the use of deep learning techniques for 3D human pose estimation has become a common approach in HRI systems. These computer vision techniques usually train a neural network from labeled images in order to estimate human pose. As a reference, some research works obtain the 3D pose estimation using a single-view RGB camera image or multi-view camera images [23,24]. These accurate methods encounter fewer problems regarding the cameras' position and calibration issues in comparison to RGB-D approaches. However, 3D human pose detection for assisted and rehabilitation robotic environments needs further improvements to achieve real-time tracking for human motion analysis with enough accuracy [25].

In our comparison, we decided to use the Kinect v2 sensor, which has a high accuracy in joint estimation while providing skeletal tracking, as described in [26]. Additionally, some research about this fact has been presented in [27], where the validity of Kinect v2 for clinical motion was compared with a MoCap system, and the error was reported to be less than 5%. We employed the skeleton obtained by two Kinect sensors as our ground truth to measure the performance of the estimation of shoulder and elbow angles using two CNN (convolutional neural network)-based human pose estimation methods in rehabilitation exercises. The selected CNN-based methods were OpenPose [28] and Detectron2 [29]. OpenPose is a multi-person pose detection system, and it can detect a total of 135 body points from a digital image [28,30]. OpenPose has been trained to produce three distinct pose models. They differ from one another in the number of estimated key points: (a) MPII is the most basic model and can estimate a total of 15 important key points: ankles, knees, hips, shoulders, elbows, wrists, necks, torsos, and head tops. (b) The COCO model is a collection of 18 points including some facial key points. (c) BODY pose provides 25 points consisting of COCO + feet keypoints [30,31]. Detectron2 was built by Facebook AI Research (FAIR) to support the rapid implementation and evaluation of novel computer vision research. Detectron2 is a ground-up rewrite of the previous Detectron version and it originates from the Mask R-CNN benchmark. Detectron2 includes high-quality implementations of state-of-the-art object detection algorithms, including DensePose [29].

## 3. Materials and Methods

The architecture of the vision system is composed of two RGBD cameras (Microsoft Kinect Xbox One, also known as Kinect v2, Microsoft, Albuquerque, NM, USA) and a webcam connected to a computer network. Each Kinect is connected to a client computer, which estimates the user skeleton joint tracking through the Microsoft Kinect Software Development Kit (SDK). Microsoft released the Kinect sensor V2 in 2013, which incorporates a RGB camera with a resolution of 1920 × 1080 pixels and a depth sensor with a resolution of 512 × 424 pixels and a working range of 5–450 cm, a 70° × 60° field of view, and a frame

rate of 15–30 fps. Data from the sensor can be accessed using the Kinect for Windows SDK 2.0, which allows tracking up to 6 users simultaneously, each with 25 joints. For each joint, the three-dimensional position is provided, as well as the orientation as quaternions. The center of the IR camera lens represents the origin of the 3D coordinate system [32,33].

The Microsoft SDK was designed for Windows platforms; therefore, rosserial is used to communicate between Windows platforms and Linux [34]. Three PCs are used for the system architecture. One of them works both as a client and server. The detailed hardware description is shown in Table 1. The RGB webcam is connected to a client PC equipped with a graphics card (GPU). Both the webcam and GPU are used by the OpenPose and Detectron2 methods for human pose estimation. The overall system topology is shown in Figure 1.

**Table 1.** Hardware setup of our system.

|  | **Server-Client** | **Client 2** | **Client 3** |
|---|---|---|---|
| OS | Ubuntu 18.04.03 Desktop (64 bit), (Canonical, London, UK) | Windows 10 Pro (64 bit), (Microsoft, Albuquerque, NM, USA) | Windows 10 Pro (64 bit) |
| Processor | Intel® Core™ i7-9750, (Intel, Santa Clara, CA, USA) | Intel® Core™ i5-8250U | Intel® Core™ i7-4700MQ |
| Memory | 16 GB | 16 GB | 16 GB |
| GPU | NVIDIA GeForce GTX 1650 GDDR5 @4 GB (128 bits), (NVIDIA, Santa Clara, CA, USA) |  |  |

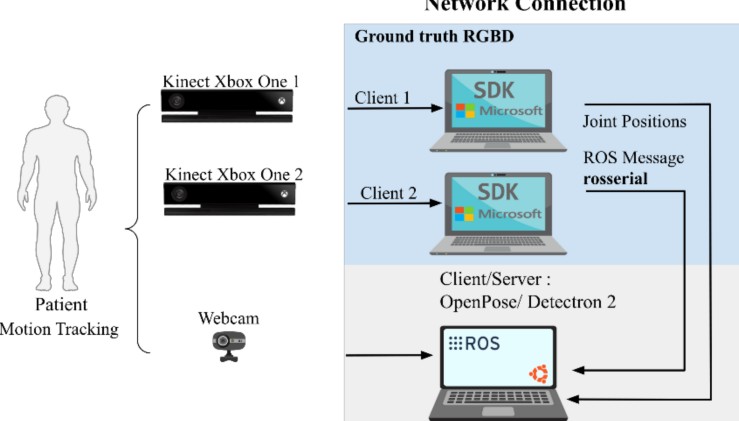

**Figure 1.** An overview of the proposed system.

*Cameras Calibration*

In order to compare the different pose estimation methods, all 3 cameras need to be calibrated in a common coordinate system. Calibration of the cameras was performed using the OpenCV multiple cameras calibration package [35]. A checkerboard pattern with known dimensions is shown so that at least two cameras can identify it at the same time. To obtain the ground truth, information of the extrinsic parameters of the cameras (translation and rotation matrix) is required, then a 3D-to-2D projection must be made in the image plane to be able to compare with the information provided from OpenPose or Detectron 2 (see Equations (1) and (2)). The acquisition and processing scheme of the data is shown in Figure 2.

$$\begin{bmatrix} X_{CK1} \\ Y_{CK1} \\ Z_{CK1} \end{bmatrix} = R \begin{bmatrix} X_{K1} \\ Y_{K1} \\ Z_{K1} \end{bmatrix} + T \tag{1}$$

$$u = \left(\frac{x}{z}\right) * f_x + c_x \quad v = \left(\frac{y}{z}\right) * f_y + c_y \tag{2}$$

where $(X_{K1}, Y_{K1}, Z_{K1})$ are the coordinates of a 3D point in the coordinate system of the Kinect 1 (IR camera), $(Xc_{K1}, Yc_{K1}, Zc_{K1})$ are the coordinates of a 3D point calculated from the Kinect 1, $T$ is the transfer vector, $R$ is the rotation matrix, $(u, v)$ are the coordinates of the projection point in pixels, $(cx, cy)$ is the image center (IR camera), and $(fx, fy)$ are the focal lengths expressed in pixel units (IR camera).

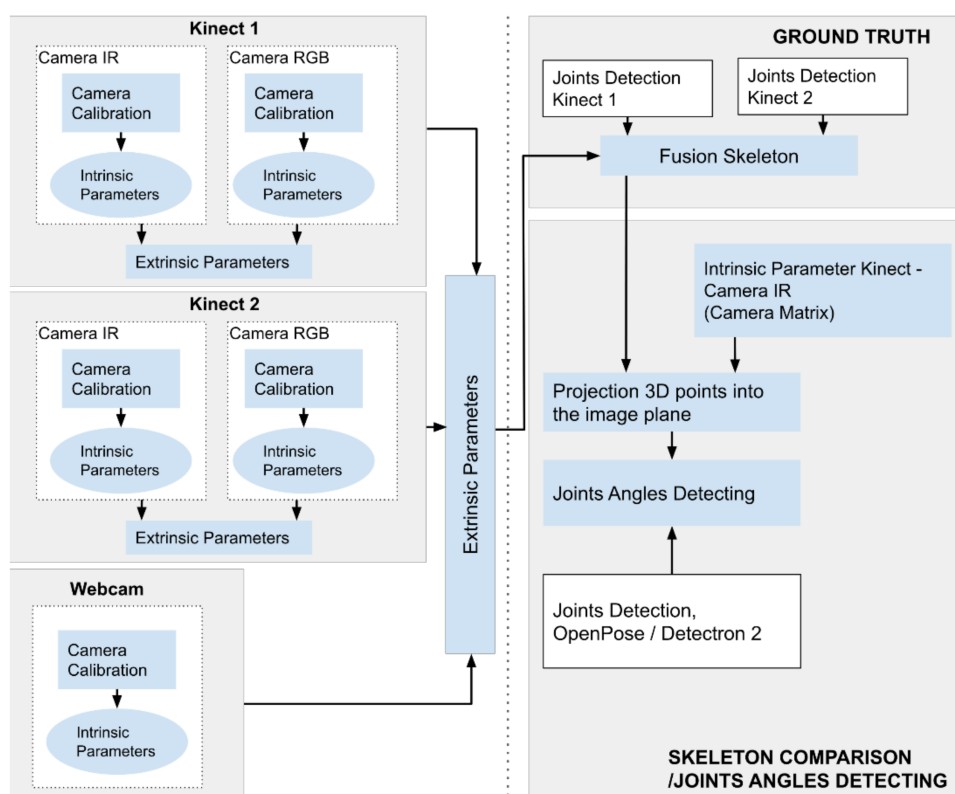

**Figure 2.** Data acquisition and processing scheme.

## 4. Experimental Setup

### 4.1. Cameras Position

The two Kinect sensors were located orthogonally to each other, as described in Figure 3a. This distribution allows the elimination of the problem of data loss caused by self-occlusion [36,37]. The distribution of the laboratory hardware setup is shown in Figure 3b. With this configuration, more precise data can be obtained on rehabilitation exercises that focus on the limbs. Finally, the webcam was located just above the Kinect 2 to reduce the errors in extrinsic parameters and to obtain a similar view with the Kinect 2.

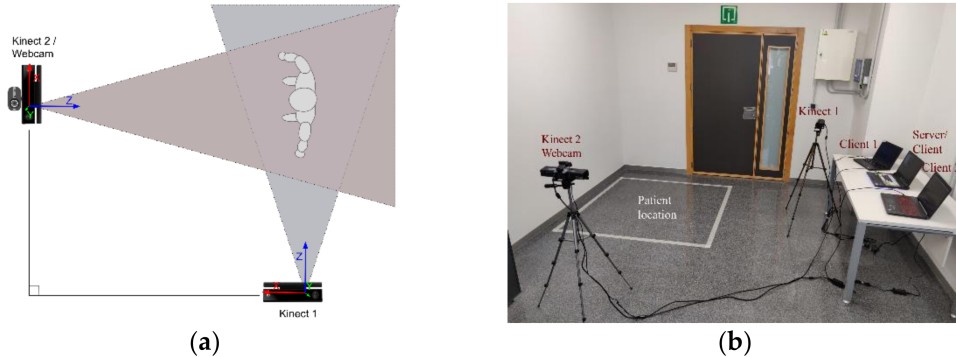

| (a) | (b) |

**Figure 3.** Our system: (**a**) Cameras distribution, (**b**) Working zone.

### 4.2. Joint Angle Measurement

The joint angle was measured as the relative angle between the longitudinal axis of two adjacent segments. These segments were composed of three points in the 2D space: a starting point, a middle point, and an end point. For the elbow joint angle, the adjacent segments were the upper arm and the forearm, respectively. Figure 4 shows the elbow and shoulder joint angles measured in this study. Let $u$ and $v$ be vectors representing two adjacent segments, where the angle between $u$ and $v$ is equal to:

$$\theta = \cos^{-1}\left(\frac{u \cdot v}{|u||v|}\right) \tag{3}$$

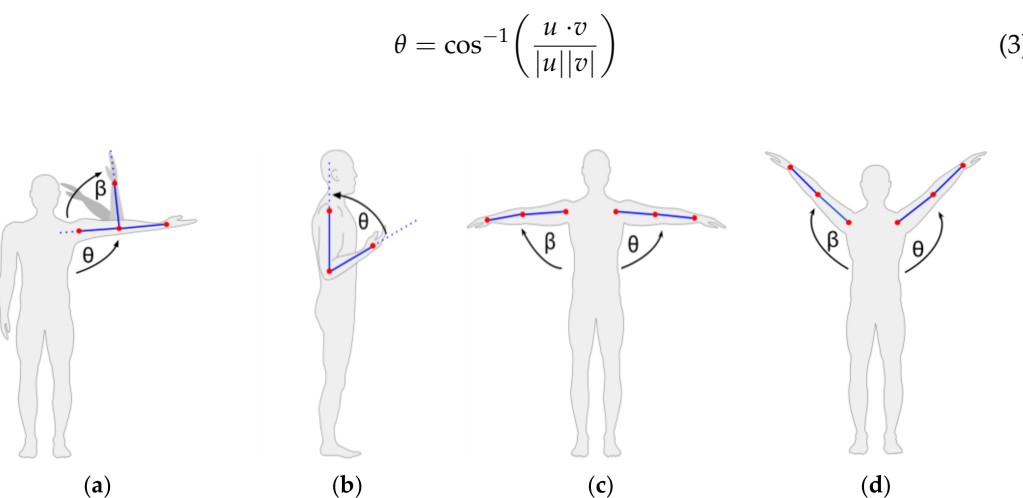

|  (a)  |  (b)  |  (c)  |  (d)  |

**Figure 4.** Rehabilitation exercises: (**a**) elbow side flexion, (**b**) elbow flexion, (**c**) shoulder extension, and (**d**) shoulder abduction.

### 4.3. Rehabilitation Exercises

Four upper limb rehabilitation exercises were proposed: elbow side flexion, elbow flexion, shoulder extension, and shoulder abduction (Figure 4). During the execution of the exercises, the cameras capture the information of the desired joints of the patient pose, and this information is used to calculate the angles θ and β obtained by the different systems.

### 4.4. Ground Truth

The ground truth of the pose estimation was calculated using the skeletons provided by the two Kinect cameras. As previously mentioned, these cameras were located at approximately 90° from each other to obtain accurate data on rehabilitation exercises that focus on the upper limb where one camera does not give fully reliable estimations. Figure 5 shows four examples of 3D human pose estimations obtained by the Kinect 2 (blue color), and the skeleton fusions (red color) during an elbow side flexion exercise. The skeleton fusion was calculated with a simple average of the Kinect 2 skeleton and the projected skeleton from the Kinect 1, which was calculated using the information obtained in the calibration phase and using Equation (3). When performing this projection, a difference is expected between the coordinates of the Kinect 2 (main camera, front view) and the Kinect 1 (auxiliary camera, side view). This difference is due to the viewing angle of each Kinect and the volume of the joints of the human.

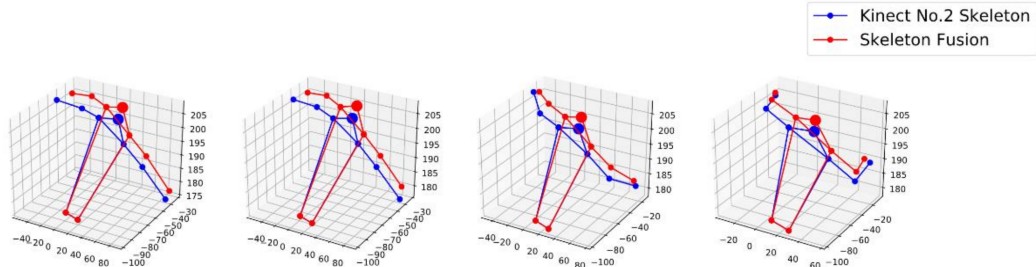

**Figure 5.** 3D skeleton poses obtained by Kinect 2 and skeleton fusion.

Figure 6 shows the X, Y, and Z positions of the left and right wrist of the two Kinects (Kinect 1 and Kinect 2) during a shoulder extension exercise and the projected points (Projected 1). As stated before, an error was expected in the calculation of the fused skeleton, and the results show how even with the fully calibrated system, we obtained some errors. For the left wrist, we measured MAEs (mean absolute errors) of 7.35, 2.16, and 3.71 cm for the coordinates X, Y, and Z, respectively. For the right wrist, we measured MAEs of 7.70, 2.90, and 3.25 cm for the coordinates X, Y, Z, respectively.

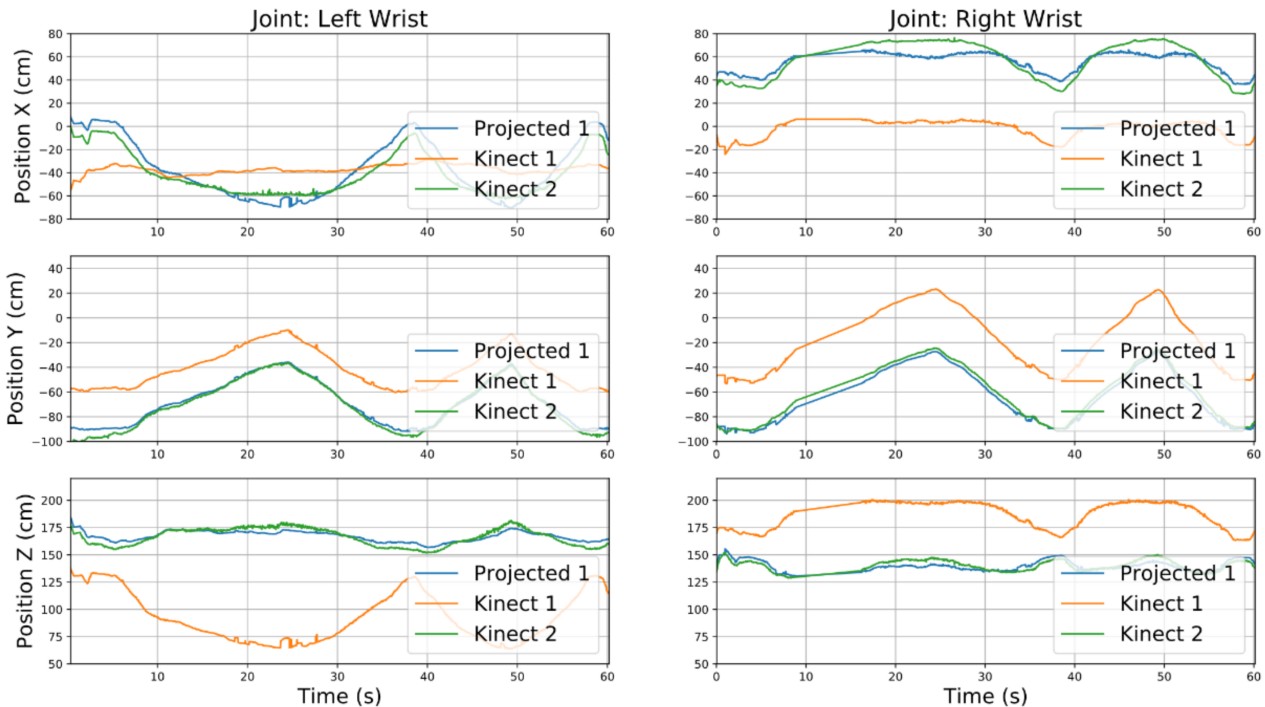

**Figure 6.** X, Y, and Z positions of left and right wrist by Kinect 1, Kinect 2, and projected position.

## 5. Experiments

The following experiments show the precision of the angles calculated using the OpenPose and Detectron 2 approaches. Only the necessary angles are shown in each experiment. The movements of exercises 1 and 2 involved only one angle, while exercises 3 and 4 involved the two aforementioned elbow and shoulder angles. A video of the experiments is available online [38]. A summary of the results of all experiments is shown in Section 6 (Results section).

### 5.1. Exercise 1: Elbow Side Flexion

The first experiment was the elbow side flexion exercise. Figure 7a shows the left elbow angle calculated with both approaches for all positions in the exercise. The left elbow angle varied between 50° and 180° with OpenPose and between 40° and 180° with

Detectron 2. For OpenPose, the root-mean-square error (RMSE) was 9.23° and the mean absolute error (MAE) was 7.53°, while for Detectron 2, the RMSE was 13.64° and the MAE was 9.32°.

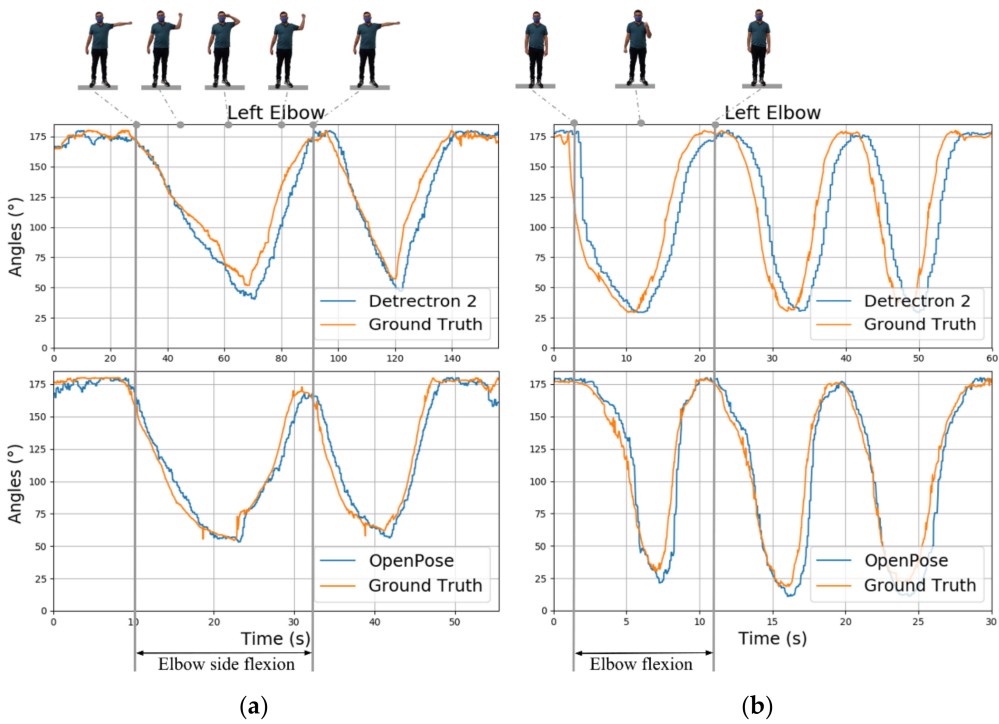

**Figure 7.** Exercise 1: elbow side flexion angles for two iterations (**a**); exercise 2: elbow flexion for three iterations (**b**).

### 5.2. Exercise 2: Elbow Flexion

The second experiment was the left elbow flexion exercise where the angle of view of the main points was not the most favorable. Figure 7b shows the results of the calculation of angles. The left elbow angle varied between 11 and 180° (i.e., straight arm) with OpenPose and between 30° and 180° with Detectron 2. For OpenPose, RMSE = 15.84° and MAE = 9.74°, while for Detectron 2, RMSE = 27.27° and MAE = 20.03°.

### 5.3. Exercise 3: Shoulder Extension

The third and fourth experiments aimed to measure the precision of both approaches when performing rehabilitation exercises with both arms in shoulder extension and abduction. Figure 8 shows the angles for the left and right shoulder in the shoulders' extension exercise. The left shoulder angle varied between 5° and 91° with OpenPose and between 4° and 98° with Detectron 2. The right shoulder angle varied between 7° and 102° with OpenPose and between 3° and 95° with Detectron 2. For the left shoulder, RMSE = 3.57° and MAE = 2.96° for OpenPose, while RMSE = 7.99° and MAE = 7.15° for Detectron 2. For the right shoulder, RMSE = 8.67° and MAE = 8.16° for OpenPose, while RMSE = 11.70° and MAE = 9.24° for Detectron 2.

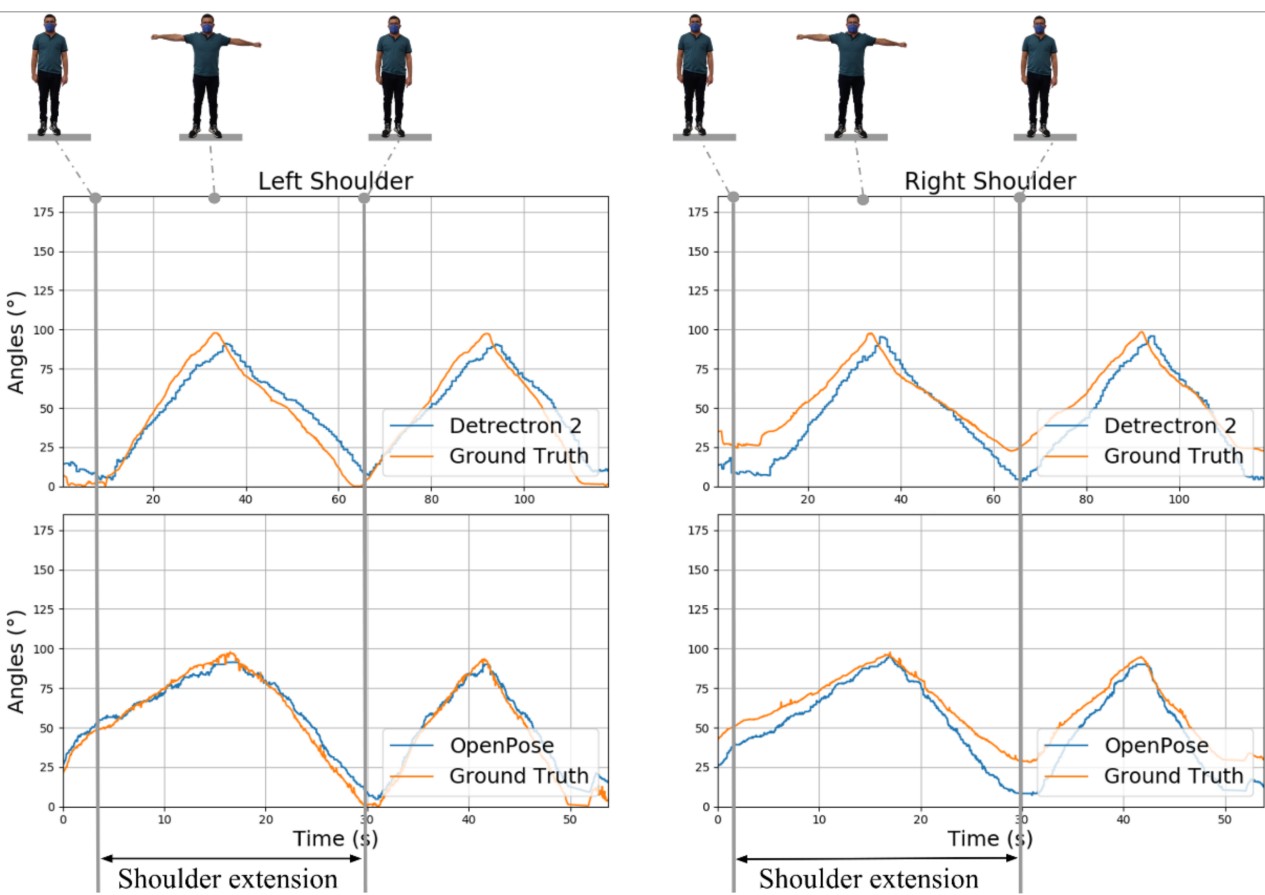

**Figure 8.** Exercise 3: angles of both shoulders in extension rehabilitation exercise for two iterations.

### 5.4. Exercise 4: Shoulder Abduction

Figure 9 shows the angles for the left shoulder and right shoulder in the shoulders' abduction exercise. The left shoulder angle varied between 80° and 128° with OpenPose and between 79° and 127° with Detectron 2. The right shoulder angle varied between 85° and 132° with OpenPose and between 80° and 138° with Detectron 2. For the left shoulder angle, RMSE = 4.68° and MAE = 4.17° for OpenPose, while RMSE = 11.79° and MAE = 7.89° for Detectron 2. For the right shoulder angle, RMSE = 4.16° and MAE = 3.74° for OpenPose, while RMSE = 12.74° and MAE = 9.03° for Detectron 2.

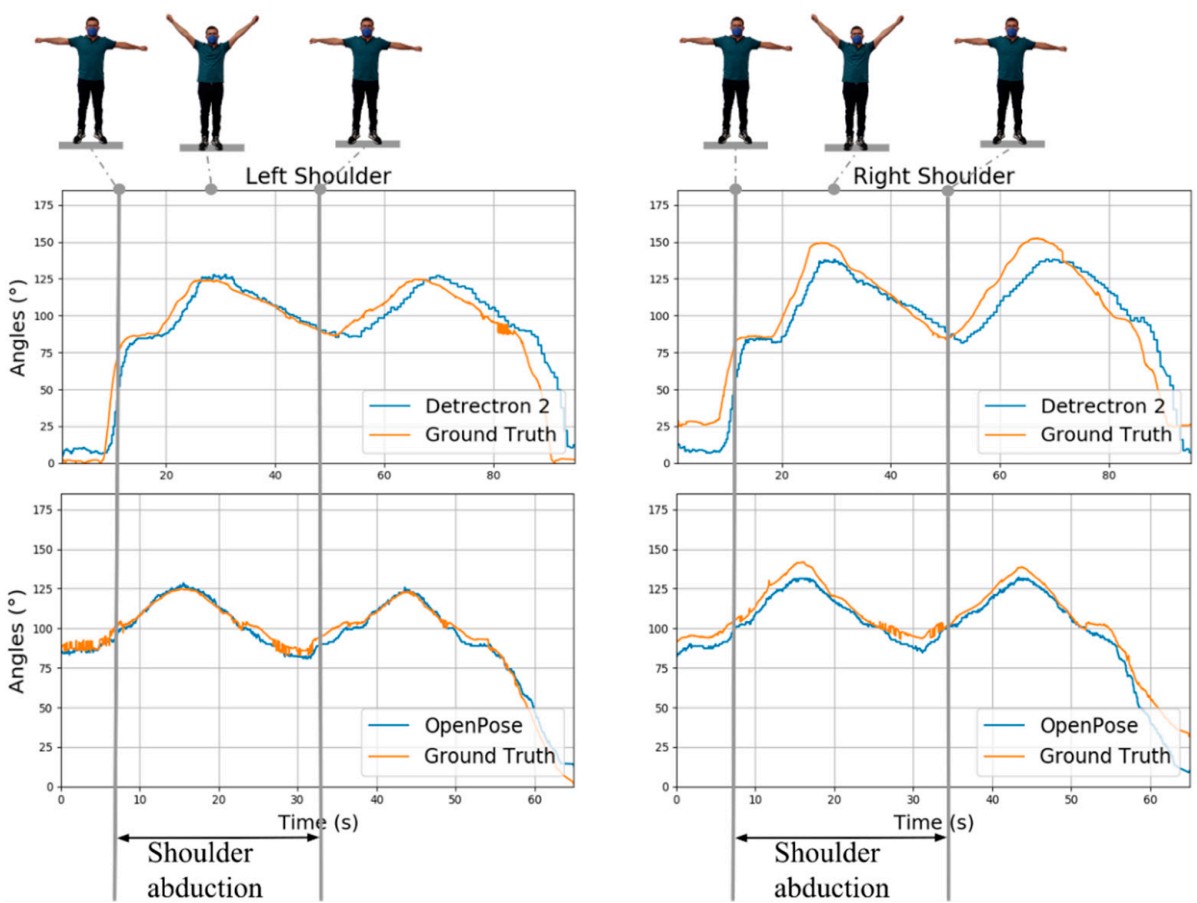

**Figure 9.** Exercise 4: Angles of both shoulders in shoulder abduction rehabilitation exercise for two iterations.

## 6. Results

Figure 10 shows a comparison of the RMSEs (root-mean-square errors) of OpenPose and Detectron 2 for each rehabilitation angle calculated in the exercises. OpenPose obtained a lower RMSE than Detectron 2 did for all four proposed rehabilitation exercises. For the exercises where the viewing angle of the webcam was not favorable, a large RMSE was obtained for both approaches. An example of these errors can be seen in the flexion left elbow exercise, which had RMSEs of 15.84° with OpenPose and 27.27° with Detectron 2. The best results were obtained by OpenPose in exercises 3 and 4 where it was easier to estimate the movement of the arms than to obtain an estimated RMSE below 5°.

To visualize the error during each exercise step, we calculated the absolute error of each method. Figure 11 shows the absolute error for each rehabilitation exercise according to both libraries. We can see again that during the exercise of flexion of the left elbow (row b), both libraries achieved high absolute errors. The results show how OpenPose had fewer error peaks, and it seemed more stable for most of the angles checked.

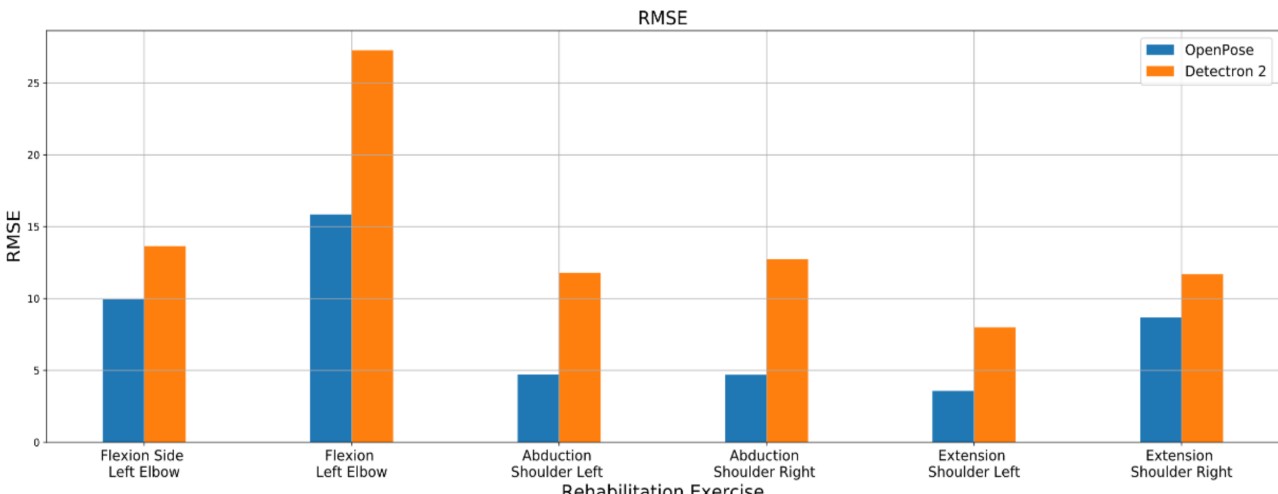

**Figure 10.** RMSE of the four rehabilitation exercises compared to ground truth.

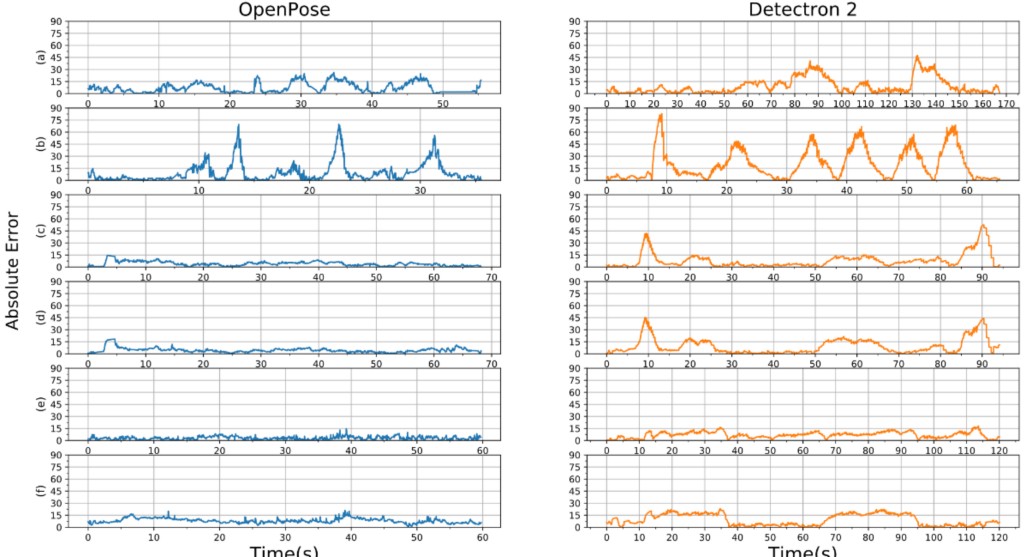

**Figure 11.** Absolute error for the four rehabilitation exercises compared to ground truth: (**a**) flexion side left elbow, (**b**) flexion left elbow, (**c**) abduction left shoulder, (**d**) abduction right shoulder, (**e**) extension left shoulder, and (**f**) extension right shoulder.

## 7. Discussion

In this article, we compared the performance of estimating shoulder and elbow angles for rehabilitation exercises using CNN-based human pose estimation methods: Open-Pose and Detectron2. Qualitatively, for the four proposed rehabilitation exercises, better results were obtained with OpenPose. OpenPose had an average RMSE of 7.9° and Detectron2 had an RMSE of 14.18°. However, for the elbow flexion exercise, which had a worse angle of view, both methods obtained high errors. According to [39], evaluation therapists tend to underestimate the range of motion by 9.41° on average for any joint movement of the upper limb. Therefore, with the results obtained in this approach, it can be concluded that OpenPose is an adequate library for evaluating patient performance in rehabilitation programs that involve the following exercises: left elbow side flexion, shoulder abduction, and shoulder extension.

Regarding the response time of the analyzed pose estimators, the performance of these methods is related directly to the available GPU. In our study, we measured a performance between 6.7 and 13 FPS with OpenPose and between 1.8 and 3 FPS with Detectron 2.

The kind of exercises related to upper limb rehabilitation is smooth and relatively slow, so the performance of OpenPose is high enough to monitor the exercises and provide useful information for rehabilitation therapy.

The major limitations of the present study were mainly related to the ground truth used; many approaches use three-dimensional motion analysis devices such as the VICON motion system or Optotrak. However, the equipment is expensive, and it requires a conditioned environment and technical skills for attaching sensors. We decided to use Kinect 2 as our ground truth because of the cost and because this sensor has a high accuracy in joint estimation while providing skeletal tracking. Another limitation of the study was that the system was only tested on a single healthy subject who participated in a single experimental session. A study with a larger group of subjects and different positions should be examined to compare the quality of the estimate of the human pose by both methods. In future works, we intend to collect data from more participants and extend this work to lower-limb movement estimations.

**Author Contributions:** Conceptualization, C.A.J., V.M. and Ó.G.H.; methodology, C.A.J., V.M. and J.L.R.; software, Ó.G.H.; validation, C.A.J., V.M., J.L.R. and Ó.G.H.; investigation, C.A.J., V.M. and Ó.G.H.; resources, C.A.J.; writing—original draft preparation, C.A.J., V.M., J.L.R. and Ó.G.H.; writing—review and editing, C.A.J., V.M., J.L.R. and Ó.G.H. All authors have read and agreed to the published version of the manuscript.

**Funding:** Óscar G. Hernández holds a grant from the Spanish Fundación Carolina, the University of Alicante, and the National Autonomous University of Honduras.

**Institutional Review Board Statement:** Not applicable.

**Informed Consent Statement:** Informed consent was obtained from all subjects involved in the study.

**Data Availability Statement:** Not applicable.

**Conflicts of Interest:** The authors declare no conflict of interest.

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
