# Peer review of "Human Pose Detection for Robotic-Assisted and Rehabilitation Environments"

_applsci, doi:10.3390/app11094183_

Round 1

Reviewer 1 Report

The paper describes camera-based upper-body human-pose detection using two Kinect cameras. Both validations in rehabilitation and robotics are missing, although appearing in the title! State of the art is not well discussed, e.g., comparable with IMU-based pose detection.

Reviewer 2 Report

The paper entitled “Human-pose detection for robotic assisted and rehabilitation environments” presents an CNN-based work about estimating shoulder and elbow angles for rehabilitation exercises. The principle and obtained results seem interesting, however there are still many issues that need to be elaborated. In addition, the grammar and sentence need to be further polished throughout the work to make it clear. Following are my comments based on the high quality of the Applied Sciences:

  1. The CNN toolbox OpenPose and Detectron 2 are widely used in computer vision based on my literature review. Please clearly state the advantages of your methods compared to others.
  2. In the principle, two Kinect(s) are used to capture the image data. For signal processing, it is important to evaluate the precision/stability of the sensing system. If they (i.e. two Kinects) are not orthogonally placed to each other, what error will you obtain and how will you detect and minimize the error? Please analyze it clearly.
  3. Please check the equation 2.
  4. Please check the unit of x axis in Fig 6. It does not make sense to do a shoulder extension exercise with 20000s (~5.5 hours).
  5. Besides, do not simply report the statistical values (e.g. MAE, RMSE, etc). Please explain the meaning behind them. For example, does it demonstrate the improvement of your proposed system? How good is it compared to existing work?
  6. Also, I am curious about the response speed of your proposed system? For example, how fast can it detect the pose change?
  7. Several sentences are grammarly incorrect and not understandable, which need to be polished. For example, Line 118, line 156, line 224, etc.

Reviewer 3 Report

Dear authors, 

thanks for the opportunity for reviewing the paper. 

I think the paper needs to be improved on some considerations: 

a) you say Human Robot Interaction at line 32, but I think you have to consider the risks about this aspect also for future

b) at line 121 you say "In our comparison, we decided to use the sensor Kinect v2, which has high accuracy in joint estimation while providing skeletal tracking" . I think is interesting in an Industry 4.0 perspectives. May you consider it for overcome your limitations? I suggest only a paper about this aspect https://doi.org/10.7232/iems.2020.19.3.551 (about a way to analyze...a structure). 

I hope you will improve soon your paper

Round 2

Reviewer 1 Report

The contribution of the work is questionable with respect to the existing state of the art in rehabilitation and robotics

Reviewer 2 Report

The author modified the paper according to the comments. And it is recommended to be published in the journal.

Reviewer 3 Report

Congratuliations!